# Obstetric Violence as an Infringement on Basic Bioethical Principles. Reflections Inspired by Focus Groups with Midwives

**DOI:** 10.3390/ijerph182312553

**Published:** 2021-11-29

**Authors:** Júlia Martín-Badia, Noemí Obregón-Gutiérrez, Josefina Goberna-Tricas

**Affiliations:** 1Department of Philosophy, University of Barcelona, 08001 Barcelona, Spain; julia.martin@ub.edu; 2University Hospital Parc Taulí, Sabadell, 08208 Barcelona, Spain; nobregon@tauli.cat; 3The Nursing Council of Barcelona, 08019 Barcelona, Spain; 4Department of Public Health, Mental Health and Perinatal Nursing, Faculty of Medicine and Health Sciences, ADHUC–Research Center for Theory, Gender and Sexuality, University of Barcelona, 08907 Barcelona, Spain

**Keywords:** obstetric violence, women, childbirth care, malpractice, bioethics, ethical aspects, midwives, humanization

## Abstract

Background: obstetric violence is still far too invisible; the word “violence” generates rejection and obstetric violence is complex to define and typify, as it is a subjective experience. It has been widely analyzed from legal, sociological, and clinical perspectives, but not equally so from the bioethical point of view. This article sets out to take a more in-depth look at the experiences of midwives in order to describe the ethical perspectives of obstetric violence. We intend to describe the effects that malpractice and violence within obstetric care have on American and European bioethical principles. Methodology: A qualitative methodology of the phenomenological tradition was used: 24 midwives participated in three focus groups. Results and Discussion: four categories were arrived at; they are “the maleficence of forgetting my vulnerability”, “beneficence requires respect for my integrity and dignity”, “my autonomy is being removed from me” and “a problem of social justice towards us, women”. Conclusion: obstetric violence infringes on the main bioethical principles (non-maleficence, beneficence, autonomy, justice, vulnerability, dignity, and integrity). Beyond whether it is called violence or not, what matters from an ethical perspective is that, as long as women have such negative experiences during pregnancy and childbirth, obstetric care needs better humanizing.

## 1. Introduction

Obstetric violence is “the appropriation of women’s bodies and reproductive processes by healthcare providers, which is expressed in a hierarchic dehumanizing treatment, an abuse of medicalization and the pathologization of natural processes, producing a loss of autonomy and free decision-making capacity in women regarding their bodies and sexuality, all of which has a negative impact in their quality of life” [1]. This phenomenon has been widely studied [2,3,4,5,6,7] with the aim of defining its main elements, which have been classified into the following categories: mistreatment or abuse, stigmatization and discrimination, violation of professional care standards, poor healthcare relationship between professionals and women, and the healthcare system’s conditions and limitations. Obstetric violence has been widely analyzed from the legal, sociological, and clinical perspectives, but not so much from the ethical point of view, which is the novelty of this work. It has not been analyzed from the perspective of North American bioethical principles (non-maleficence, beneficence, autonomy, and justice), nor from that of European ones (vulnerability, integrity, and dignity) [8,9] (see Table 1).

From the bioethical perspective, the first reflection that needs to be taken is around the concept of obstetric violence itself. Although the United Nations (UN) states that obstetric violence is a human rights violation [10] and that in 2019 the Council of Europe passed a motion on obstetric and gynecological violence [11], the World Health Organization (WHO) calls for action against the “disrespectful and abusive treatment [experienced by women] during childbirth”, without using the term “obstetric violence” [12]. This shows to which extent the word “violence” generates controversy. For some, it contributes to the visualization of adverse experiences in childbirth and to punish those professionals who have generated them, in order to improve the quality of healthcare [13]. For others, the solution should be to prevent these experiences from happening, instead of punishing after the fact [14] (pp. 93–104)—which could generate defensive medicine exerted by professionals afraid of being sued. The latter would prefer concepts closer to the WHO’s expression, such as “malpractice”, “disrespectful care”, or “dehumanized care”.

The question, then, arises: is obstetric violence an appropriate concept? To begin with, it is very wide and subjective. In order to make it precise, the word “violence” needs to be defined. It is not the aim of this paper to theorize about this, but it seems pertinent to mention some of the most usual definitions. According to Aristotle, a violent action is one that goes against its own natural path. In this sense, the medicalization (or at least over-medicalization) of a natural process such as childbirth is a violent act. From a political perspective, violence refers to the exercise of non-legitimate power [15]. Healthcare professionals have power over their patients as long as they have technical skills. The question is whether using the power that the technical skills entail is legitimate or not. This exercise of power is not self-legitimized, but legitimized by patients, only if there is a trusting relationship between them both, in which the patient consents to the use of the professional’s technical skills and the latter uses these skills to take care of the patient and not to harm him or her. The legitimacy, then, of the professional’s use of power should be assessed in every medical action and, therefore, the appropriateness of the concept of “violence” should be assessed in every woman’s bad experience. Beyond the concept, what cannot be denied is that women have bad experiences during delivery, judging by the posttraumatic stress disorder suffered by many of them [16]. Therefore, a humanized relationship and an appropriate and trusting communication between women and professionals is needed in order to reach a mutual comprehension and recognition about women’s feelings and professionals’ reasons to act the way they do. From this standpoint, it may still be difficult to define violence or malpractice, but it may be easier to define good practice, that is to say, the practice that both the provider and the receiver find appropriate as based on scientific evidence, clinical experience, and patients’ values. In this paper we offer a bioethical analysis of obstetric violence from midwives’ narrated experiences, as they are the referral healthcare professionals in childbirth and, therefore, the ones with whom women need to build this trusting relationship that defines good practice. 

The core objective of this paper is to describe the ethical perspective of obstetric violence through the analysis of Catalan midwives’ experiences. First, we have tried to understand and take an in-depth look at the experiences of midwives in order to find out their experience in obstetric violence, and, subsequently, we have delved into the ethical implications of obstetric violence both from the perspective of the American bioethical principles (non-maleficence, beneficence, autonomy, and justice) as from the European ones (vulnerability, integrity, and dignity).

## 2. Methods

### 2.1. Design

We used a qualitative methodology based on Husserl’s phenomenology, as a qualitative methodology allows naturalistic interpretations and approaches to the research subject. According to Husserl, phenomenology is “a paradigm that seeks to explain the nature of things, the essence and the veracity of phenomena. The objective that it pursues is to understand the experience lived in its complexity […] and the meanings centered on the phenomenon” [17].

Focus group discussions were used as a technique in order to allow participants to share their experiences through open and freely fluent discussions. These experiences, seen as a unit of analysis, provide “a way of interpreting, assessing and making sense of reality, whilst reflecting the unity of socio-cultural and personal aspects” [18]. 

### 2.2. Context of the Study

The study was carried out in Spain, specifically, in the Autonomous Community of Catalonia (province of Barcelona). Spain is divided politically into seventeen Autonomous Communities which have full powers on healthcare matters. Catalonia is situated in the northeast of the Iberian Peninsula. About 30% of Catalonia’s population live within the administrative limits of the city of Barcelona, which itself is contained in the Barcelona metropolitan area. Highly complex hospitals located in Barcelona are the reference centers for pathologies existing around Catalonia. In the rest of the province of Barcelona there are several regional hospitals of medium complexity. The midwives who participated in the study work in five hospitals of different complexities (three high complexity hospitals and two regional or medium complexity hospitals), as well as in primary care (See Table 2). 

In Catalonia, midwives are the first point of contact for pregnant women within the health system, as they monitor and control a normal pregnancy and attend normal births in hospitals. In Spain, hospital deliveries constitute the majority of deliveries; delivery centers and home deliveries are anecdotal.

### 2.3. Selection and Characteristics of Participants

Three focus groups were held with active working midwives. Some theoretical selection criteria were established, and these were used to cover the largest possible number of midwife profiles. For the selection of the participants, we have the collaboration of the Nursing Council of Barcelona (in Spain, midwives must perform the specialty of obstetric and gynecological nursing after having graduated in nursing and are required to be registered in the Nursing Council for the active exercise of the profession). Through the Council’s records, we were able to identify the list of midwives in the province of Barcelona who have been in active practice for more than a year and who meet the following criteria:Midwives who work in Level II Hospitals (regional hospitals of medium complexity);Midwives who work in Level III Hospitals (hospitals with a high level of complexity);Midwives working in Primary Care.

Those midwives who worked in other services other than care for pregnancy or childbirth or with less than one year of professional practice were excluded.

Two researchers (NOG and JGT) contacted via email those midwives who met the specified criteria and who wanted to participate in the focus groups and agreed upon the date of participation with them. A total of 24 midwives—23 women and one man—participated in the three focus group discussions. Their profiles are shown in Table 2 (the names have been changed to preserve the anonymity and confidentiality of the participants).

An interview script was used to conduct the focus groups. In the first place, participants were asked “What does the concept of Obstetric Violence suggest to you?” After this first stimulus-question the discussions continued with an open dialogue guided by the previously elaborated script (See Table 3).

Focus groups were held during March 2018. Focus groups discussion were conducted by the three researchers (JGT and JMB in groups a and b; and NOG and JMB in group c) in the official headquarters of the Nursing Council of Barcelona. There were large rooms with chairs arranged in a circle specifically for this purpose. The duration of the focus groups ranged from 110 to 150 min.

### 2.4. Ethical Aspects

The research was approved by the University of Barcelona Bioethics Committee (IRB00003099). The objective and ethical considerations of the study were explained to all the participants by e-mail, and they were also sent the information and an informed consent form by email, which they signed and returned to the Main Researcher the same day of the focus groups’ discussion.

### 2.5. Criteria for Methodological Rigor

We considered the list of questions contained in the Standards for Reporting Qualitative Research (SRQR) throughout the development of the study and in the drafting of the final report. Furthermore, we used the following quality criteria in accordance with Calderón [19]: (a) “epistemological adequacy”, that is to say, reviewing the formulation of the research question and the coherence of the process; (b) “relevance”, the need to know the experiences of midwives about obstetric violence and the ethical aspects about it. Similarly, the (c) “validity” criterion is not intended to be understood in terms of statistical probability but rather in terms of relevance and interpretivism, and so, an appropriate process has been sought for the selection of participants and in order to guarantee rigor in the analysis in order to understand meaning and look for in-depth generalizable explanations from a logical point of view that are transferable according to the contextual circumstances in which the research was carried out. In our case, we strived for maximum variability in each of the groups, both in terms of workplace (high complexity hospital, medium complexity, or primary care), and in years of experience. Finally, (d) “reflexivity”, that is, it is also important to recognize the position of the researchers both as midwives and researchers because some of them are also immersed in the scenario of childbirth attention and obstetric violence.

### 2.6. Data Analysis

The three focus group discussions were recorded in MP3 format and subsequently transcribed by two Researchers (JGT and JMB) and a collaborator who was external to the research. They were then analyzed using qualitative research methods based on the criteria of Taylor and Bogdan [20].

The first step consisted of a careful reading of the transcriptions to obtain ideas and intuitions. In this phase, after several re-readings, we identified those aspects that the midwives explained, which were clearly related to bioethical principles. Two researchers (JMB and JGT) independently identified the initial ideas before sharing their observations and agreeing on their applications. The second step involved categorizing the data into information units and grouping them into categories based on similarity, which responded to the objectives of the study. The same two researchers (JMB and JGT) independently identified the initial codes and then shared and agreed upon them. The researchers analyzed whether and how obstetric violence fails to respect the main principles of biomedical ethics that healthcare professionals must follow in their daily practice, both the American ones [8] and the European ones [9]. 

The third step consisted of relativizing the data to contextualize them [20]. The codes and categories that emerged were discussed by all the members of the research team.

## 3. Results and Discussion

After the analysis of the transcriptions, four categories were identified as shown in Table 4.

### 3.1. The Maleficence of Forgetting Women’s Vulnerability

Healthcare professionals are subjected to the obligation of “first doing no harm” to their patients, as the Hippocratic Oath states. It may not be possible to help a severely ill patient, but at least he or she should not be harmed [8].

In gynecological care, women share their physical intimacy with professionals as they show their naked bodies. According to midwives, laboring women allow physicians to “work with their bodies” during processes—pregnancy and childbirth—that have a high emotional and psychological impact, as well as a cultural dimension, so it becomes necessary to have a trusting and close relationship with those professionals:
*“You are working with a vital process with the woman’s body and, well, it seems to me a paternalistic attitude, the fact of imposing on her body and a process that should be something normal and logical, and so, and many times we forget, I think that many times we forget the cultural part, the religious part, and so, sometimes we forget about it completely.”*(Rachel, group C).

There is often a reifying gaze from professionals to women’s laboring bodies that transforms them into dirty, overly sexual, insufficiently feminine, and dysfunctional bodies, which generates so-called “gendered shame” on those women [21].

Assuming that this is malpractice, some midwives argue that maleficence or malpractice is just one form of violence, as violence is not always exerted in a brutal manner, but can be a hidden violence. Others argue that there is a difference between how an action is experienced (as violent or not) and how professionals work (good or malpractice):
*“People associate violence to something brutal; I have seen people who…in an underhanded way exert a violence which is not like the visible one, and this is the most dangerous thing in the world.”*(Celine, group A)
*“One thing is the feeling that you give to an act that is done, and another thing is the professional practice with which one works in a center.”*(Barbara, group C)

As suggested before, the limits between violence and malpractice are not clear [4,22,23]. 

Beyond the concept of obstetric violence, what matters is that, according to midwives, during childbirth women experience “aggression”, “harm”, or “mistreatment” that can be physical, verbal, emotional, psychological, or moral [24]:
*“Aggressions that women suffer within the healthcare environment, and by aggression I mean verbal, physical, and emotional.”*(Sara, group A)

In addition, this malpractice can be produced by professionals’ acts, omissions (i.e., not informing, not obtaining consent, or not stopping colleagues’ malpractice) or overacting (i.e., medicalization or over-interventionism in natural processes) [25]. Midwives agree:
*“In the end, we also do things by omission. One thing is performing the action, and another is the omission, when we tear our eyes away from it, we also do it unintentionally.”*(Alisa, group B)

An interesting debate that emerged in the focus groups with the midwives is whether the professional should always be conscious of their maleficent acts in order for them to be considered violent. Some midwives argue that without conscience there is no violence:
*“Violence, I think, is this; it is acting...consciously that you are doing it and you know that it will cause a prejudice to the woman, whatever you do.”*(Hugh, group C)

This statement is contradictory with the idea that obstetric violence is subjective, as it depends on women’s perception. Without intending to solve this debate, from the authors’ point of view, there can be an agreement in two facts. On the one hand, whenever the professionals’ acts or omissions entail bad consequences (women have bad experiences) there has been malpractice, no matter whether the professional was conscious of it or not. On the other hand, whenever the professional is consciously acting inappropriately (i.e., because they lack communication skills or because of working in the environment of care-providing under pressure), there is also malpractice, independently of whether a given woman does not experience it badly. The debate should be whether maltpractice is a synonym of violence, a low degree of it or if they are two different concepts.

Ultimately, then, if there is consciousness from the professional or bad experience from a woman, there is, at least, malpractice and maybe even violence. Regardless of the relationship between violence and malpractice, both constitute an infringement on the non-maleficence principle through taking advantage of, instead of respecting and trying to revert, women’s vulnerability, which in turn contributes to increase their vulnerability.

### 3.2. Beneficence Requires Respect for Women’s Integrity and Dignity

According to Beauchamp and Childress [8], beneficence comes after non-maleficence: the second most important mandate any healthcare professional has is to provide the maximum benefit or the highest possible level of well-being to their patients. Ensuring patients’ well-being requires shifting from a biocentric conception of medicine, according to which only the body is considered, to a biopsychosocial conception of health [26], that leads to providing care that is respectful of the patients’ integrity and dignity [9].

In the context of childbirth, this has also been defined as a “respectful maternity care” [27], which is often harmed as childbirth is seen as a medical procedure and not as a natural process. Consequently, women are not treated as persons, but as objects, so they experience a dehumanized care that affects their dignity and depersonalizes them [24]. According to midwives, this dehumanized care entails abuse, disrespect, humiliation, tactless attitudes, a lack of education, a lack of intimacy (i.e., leaving doors open in labor rooms), infantilization, and even therapeutic persistence:
*“It suggests to me an abuse to integrity…”*(Carol, group B)
*“Acts of humiliation, lack of respect…”*(Cameron, group C)
*“Sometimes the words we use like ‘mummy’ and ‘daddy’, these kinds of things that are used that are totally inappropriate and that infantilize the woman and make no sense.”*(Amy, group B)
*“I associate it with therapeutic cruelty.”*(Hugh, group C)
*“As for that WHO says, which defines it as a lack of respect and considers it as a public health issue, I totally agree.”*(Sara, group A)

This shows a biocentric conception of medicine, due to which only the woman’s body is considered, not the woman as a biopsychosocial being. This lack of consideration for women’s dignity and integrated well-being is a lack of humanization that can affect the childbirth’s positive outcome [28], which is the greatest benefit that pregnant women expect to receive when they put themselves in the hands of professionals. This is why obstetric violence infringes upon the beneficence principle.

### 3.3. Women’s Autonomy Is Being Removed from Them

As stated before, in the context of obstetric violence women’s vulnerability is infringed upon. Vulnerability can be defined as the lack of resources or sufficient autonomy to transform these into well-being [29]. The principle of autonomy can be defined as the respect for decisions made by competent patients [8], which requires providing them with accurate and understandable information that they can use in order to provide themselves with well-being.

According to midwives, there is a paradox in today’s society. On the one hand, women are increasingly empowered, willing to make their voices heard and with a higher critical capability to discriminate between good and malpractice in order to choose what they prefer. On the other hand, the healthcare system is not fully aware of it and, through the professionals’ abuse of authority, keeps undermining women’s rights, freedom, decision-making capacity and, ultimately, their autonomy [30]:
*“I think we are at a turning point, because women have a higher level of instruction, they know more, are better informed…society is changing, women want to empower themselves, they want to do things and are asking us to do so”.*(Alisa, group B)
*“Women have their rights, authority, autonomy, and empowerment taken away. They are left defenseless because you have the truth. This is an abuse of authority”.*(Malory, group C)

According to midwives, there has been an epistemological loss, as, traditionally, in home birth, women knew how to give birth, but now they have lost this wisdom, as the hospitalization and medicalization of childbirth has made them lose their knowledge and power. This is the reason why maternal education emerged:
*“Childbirth was something natural and you gave birth and there was not even…; women, my mother for instance, maternal education is a nonsense, you didn’t need to learn how to breast-feed your baby nor how to give birth, because everybody did it and all women were prepared for it, they use to give birth at home and that was OK.”*(Barbara, group C)

This is supported by the literature, which states that, when it comes to childbirth, women should be the first authorities in the field as it is about their bodies and health [31].

In this sense, midwives feel that women are re-empowering themselves again:
*“I think now women are beginning to be listened to, because they are beginning to mobilize, from the very moment when they ask for a natural delivery, they already ask again for home birth. Then, they are given a more important role that, let’s say, the hospitalization of childbirth took away from them.”*(Hugh, group C)

Within this need for empowerment, women are asking more and more for a “normal birth” (birth with the minimum medical intervention), in order to regain their authority, as they think their bodies are exceptionally wise [32]. Even home birth is being asked for again [33], in order to avoid obstetric violence [34].

However, there is still a widespread undermining of women’s autonomy, as midwives state: there are poor or missing informed consent processes, because sometimes professionals have poor communication skills (it is not about providing long explanations, but about ensuring that women understand what they are told), they provide biased or manipulated information (in order to direct women’s answers), and there is even a lack of information:
*“You are informing the woman, but maybe the information is directed, isn’t it?…and then the information we give, the consent we obtain, is not real. Because you are telling something to this woman whom you are directing”.*(Malory, group C)

According to Kingma [35] and Jolly et al. [36], today the importance of informed consent is at risk of being denied or disregarded in maternity care, where women still claim to be the decision-makers.

In order to avoid this, midwives say, decisions should be accompanied both by professionals and by people of the woman’s choosing (partner or family). Unfortunately, this accompaniment is not always appropriately done:
*“To me, violence is also the physical space…the couple’s intimacy, who do you want to be accompanied by? Whom do you want to stay with?”*(Fiona, group A)
*As a general rule, I think we are not aware, as professionals, of the quantity and quality of accompaniment we can offer. We are not accompanying women…”*(Francesca, Group A)

Appropriately accompanying women during delivery means that authority should be, at least, shared by women and professionals [37,38], as stated by the European version of the autonomy principle according to which autonomy is relational, not individualistic [9]. It implies that respecting others’ autonomy is not only a matter of not hampering their decisions (the American-individualistic version of the principle), but it is also a matter of accompanying decision-making processes. That is according to Zamani et al. [39], who say that social support contributes to helping women find the best path in which to have their wishes and preferences respected in a safe way. However, it has been argued that obstetric violence demolishes relationships and interdependence [40].

Ultimately, midwives suggest that, instead of supporting women’s empowerment needs by respecting their autonomy and authority, the healthcare system has a patronizing attitude towards them [41]:
*“I think it scares us, because we come from a healthcare system, paternalistic and sexist, in which we are used to making decisions for others…for us it is much easier to believe that what we are doing is what happens most often, and we come from the idea that we possess the knowledge and that the other person does not, so we are in this situation of power: I do know and you don’t know. And it’s like this, I think, it scares us to know that we don’t know what we know as much as we think, and that, in the end, what we’re affecting is a person who has the right to decide what’s best for her and not for us.”*(Alisa, group B)

For all this, it can be argued that obstetric violence infringes on the autonomy principle, in both its American and European versions, as women lack both appropriate information and appropriate accompaniment to use it to ensure a birth that respects their own values and preferences.

### 3.4. A problem of Social Justice towards Women

The definition of the American bioethical principle of justice [9] says that it has to do with an appropriate distribution of resources, which is not an egalitarian one (giving everybody the same), but an equitable one (giving to each according to their needs). In relation to this, social justice is the equitable distribution of opportunities to develop capacities, exercise rights, and reach self-realization [42].

According to the analysis of the midwives’ discourse, three main reflections can be drawn regarding the management of the principle of justice in childbirth.

The first one is that obstetric violence is related to healthcare violence and to violence towards women. Midwives state that obstetric violence is framed in healthcare violence that is suffered by many users of healthcare services and that this structural violence towards women starts far before pregnancy and childbirth:
*“Yes, it is mixed, it has the healthcare component, and, in addition, she is a woman.”*(Alisa, group B)
*“I don’t like to call it obstetric violence because I think that at the end, I dread it being restricted to obstetrics, I think it is violence towards all women…not only at this moment (childbirth), there are other moments like contraception, sexuality, and adolescence in which we are also a bit marginalized…*”(Araminta, group B)

The literature supports that obstetric violence is an added violence that the healthcare system inflicts on women for the very fact of being women [43,44,45], as gender is a social determinant of Health [46]. It seems clear that the biocentric perception of patients that is still rooted in our healthcare system carries an androcentric perspective of illness [47] (p. 12). It is also supported that there is general violence towards women [48], that it is not only physical [49], but also psychological and moral [50]. 

The second reflection that arises from the midwives’ discourse is that social inequality leads to obstetric vulnerability. According to Martín-Badia [51], there is an inequality between gynecologists and midwives, the latter having less authority than the former:
*“I felt like going to work in new places and having to, like having to demonstrate that I know how to assist childbirth, you know?…Well, now when you become more confident with the “gynes” (gynecologists), they allow you to do more, but for me at the beginning, maybe it was my perception, okay, but I felt like this, like I had to demonstrate…”*(Rachel, group C)

This patriarchal relationship between professionals is framed in a society that does not equally treat women and men as the social expectations on them are not the same [52]. According to midwives, women are expected to be mothers:
*“…people get married, buy a dog, a house and the last thing is the child, it is like the same process…the objective is a child, what is there in between if we could obviate it, most would obviate it, wouldn’t they? This is my feeling.”*(Rita, group C)

In this sense, midwives theorize about two versions of feminism that attack each other within the struggle for establishing which one better defends women’s rights. One version defends maternity as a legitimate desire that should not lead to considering women who are willing to be mothers as legitimizers of gender roles. The other version defends career progress as a right that should be respected in those women who place it above maternity:
*“If we talk about the versions of feminism, the one which defends maternity and the one which defends career progress; from our own feminism we attack the other one.”*(Eleonor, group B)

The consequence, midwives say, is that women have an internal fight between the desire of becoming mothers and the culpability of having to leave their job for some time, which often implies not being able to progress in their careers as much as men:
*“…you feel that it is right that you set your professionality, the work world a bit aside and on the one hand you feel good, because you want to raise your child, but…you also feel a bit guilty, like I if I do this I cannot do the other thing, it is a bit hard to be able to do both things or to be so involved in everything, it is like the internal fight about what you want…”*(Carol, group B)

This situation shows that, socially, men are still above women. Midwives say that women introduced themselves in the world of work in order to be able to self-actualize at the same level as men. Our society has allowed it, but without considering maternity as an added value and by asking women to demonstrate their skills, whereas men are supposed to have them:
*“Women have been incorporating ourselves into the labor market by looking at male roles, by wanting to imitate the same situation, that is to say, I want to have the same education, the same working day…nobody thinks of a woman that has two or three children: if she is capable of organizing her home, she will probably have more organizational skills in a job than another person…I think we are at a point in which when men are in managing positions women have to give 120%, I think women have to demonstrate that we can.”*(Araminta, group B)

Obstetric vulnerability arises from the fact that women are socially seen as vulnerable, a concept that is related to weakness and dependency [53]. This inequality between men and women is evident in the work field [54], among others. 

The third reflection that arose from the midwives’ discourse is that women’s socio-cultural context determines their expectations about childbirth and, therefore, their experiences. According to midwives, women usually have very high expectations, which generate a great frustration when the birthing process does not go as planned:
*“Sometimes they have a great frustration, because they think that medicine is like math, that two plus two equals four, that everything is squared, that there is evidence about everything and when that does not end well or they do not get what they had thought, excellence is not accomplished. The woman has very high expectations.”*(Fiona, Group A)

Women’s high expectations, as midwives say, come from the fact that women, at least in the West, are used to living well and therefore think that childbirth must be easy, quick, and painless:
*“We have a woman that generally…maybe she has the Thermomix at home and the robot cooks for her, obviously uses a washing machine…the car can self-park…whose life is very easy in general. Now childbirth, pregnancy, has nothing to do with it, it won’t be either easy or quick, it will be painful, what we usually try to avoid…we try to liberate ourselves from pain and from anything that annoys us”.*(Francesca, group A)

This reflection is according to some authors [55,56,57]. The literature supports that women have a fear of childbirth when they associate it with pain [58]. In reality, Spain is the country of the “epidural analgesic epidemic” [59], because our society wants to avoid pain, so we try to free ourselves from anything that bothers us.

However, as stated by midwives, it is not a matter of being a western woman with a high education level or an eastern woman without academic training, but about how easy one’s life is, about one’s values and capacity to control pain, and about one’s responsibility (and one would add capabilities):
*“Well, you can be a woman that does not think about maternity…but, well, the day comes, and you are pregnant,…and what do you do? Well, you get informed about what pregnancy means, and you enter this world and the information, I insist, as a responsible person.”*(Naomi, Group A)

The three previous reflections on social justice show that women are discriminated against in relation to other patients, as gender is a social determinant of health; that they are also discriminated in relation to men, as women are expected to be mothers at the expense of their professional career and seen as more weak and vulnerable, and less skilled; and that there are different expectations about childbirth between women from different social contexts, which leads to different experiences of childbirth. This demonstrates that both the commission and the experience of obstetric violence have to do with a lack of social justice or, to put it in other words, that both are facilitated by our society’s infringement on the principle of justice.

## 4. Limitations and Future Research Lines

This study has some limitations. The first one is that its qualitative nature allows an in-depth analysis about a topic, in this case the ethical perspective of obstetric violence, but it impedes generalizing the findings. Another limitation is that only one man was included in the focus groups, due to the fact that there are few male midwives. The last limitation is that the three focus groups with midwives took place before the irruption of the COVID-19 pandemic. Were it carried out now, the results would be different. Therefore, a future line of research could be to analyze the impact of the pandemic on obstetric violence, given that the International Confederation of Midwives stresses that women- and midwives’ rights have been infringed on in many countries by the introduction of protocols that are inappropriate, not evidence-based, and harmful for women and their babies [60]. 

## 5. Conclusions

In this paper it has been analyzed how, according to midwives, obstetric violence infringes on the basic principles of bioethics, both the American ones (non-maleficence, beneficence, autonomy, and justice) and the European ones (vulnerability, integrity and dignity), which should not be seen as compartmentalized, but as inextricably linked.

From an individual perspective, the conclusion is that women are victims of malpractice (infringement on the non-maleficence principle) as long as their vulnerability is not considered, and, consequently, it is increased (infringement on the principle of attending vulnerability). Women are not seen as biopsychosocial beings, so their dignity is frequently damaged and ultimately their well-being is diminished (infringement on the integrity, dignity, and beneficence principles). They are not fully aware of their rights and their decision-making capacity is not promoted (infringement on the autonomy principle).

From a social perspective, the conclusion is that women suffer from inequalities not only in obstetric care, but also in the healthcare system as well as societally. Nowadays, the healthcare system is sometimes still biocentric and paternalistic towards all patients and it is also androcentric as men are the measure of illnesses. Women are thus doubly victimized in it: because of being patients and because of being women. In addition, violence in obstetric care is framed in violence towards women, which begins far before pregnancy, in adolescence. Violence towards women is a social and political problem rooted in the patriarchal nature of our society that allows inequalities between men and women. These inequalities together with their sociocultural context determine women’s experiences within childbirth care. All this demonstrates that both the commission and the experience of obstetric violence have to do with a lack of justice, that is to say, both are facilitated by our society’s infringement on the principle of justice.

Finally, regarding the suitability of the concept of obstetric violence, what matters from an ethical point of view is recognizing that women have bad experiences during childbirth, which means that in obstetric care not everything is being done properly. Whether these bad experiences should be called violence, malpractice, or anything else should be a concern only as long as it affects the professionals’ readiness to recognize that they do not always act in an appropriate way and to actively try to change it. Labels do not solve the problem, on the contrary, they can aggravate it if they generate defensive attitudes that prevent one from recognizing one’s mistakes. Even if labels make these mistakes and experiences visible, visibility cannot be the goal, but must be the means to define and implement strategies to continue to humanize obstetric care. We may stop labeling, and start thinking and acting, so that *Homo Categoricus* (in a Kantian sense) does not extinguish *Homo Sapiens*.

## Figures and Tables

**Table 1 ijerph-18-12553-t001:** North American and European biomedical ethical principles.

Four Principles of North American Biomedical Ethics	Four Principles of European Bioethics
Autonomy (individualistic sense): we must respect decisions made by competent patients and protect those patients who lack decision-making capacity (i.e., minors, disabled patients, patients in coma…).	Autonomy (relational sense): we must take the patients’ social context into account, as every person lives immersed in a network of relationships that affect and are affected by the decisions she makes.
Beneficence: we should ensure the maximum possible benefit or well-being for each patient by taking what s/he considers good for him- or herself into account, which usually has to do with being able to manage his/her daily life and pursuing his/her life projects.	Vulnerability: we must be aware that, despite the fact that all humans are essentially vulnerable, patients are especially so, as illness makes them fragile, and, at the same time, it is a threat that makes them dependent on professionals.
Non-maleficence (primum non nocere): even if we cannot increase the patient’s benefit or well-being, we must try not to harm them by avoiding unnecessary or disproportionate risks.	Integrity: we need to understand the patient as a biopsychosocial being, that is, as a person with many dimensions as a well as personal values, beliefs, and preferences. We must understand that well-being is physical, mental, social, and spiritual.
Justice: we must distribute resources with equity (according to individual needs), not in an egalitarian way (giving the same to everyone).	Dignity: we must respect the inherent value that every human being has for the mere fact of being a human being. Dignity is not lost even when autonomy is.

**Table 2 ijerph-18-12553-t002:** Profile of participating midwives.

**Focus Group A**		
**Name**	**Years Working**	**Hospital/Primary Care**
Sara	8	Both
Fiona	40	Both
Margaret	18	Hospital (level II)
Mary	11	Hospital (level II)
Celine	25	Hospital (level III)
Franchesca	4	Primary Care
Naomi	20	Hospital (level II)
**Focus Group B**		
**Name**	**Years Working**	**Hospital/Primary Care**
Carol	5	Primary care
Maggie	31	Hospital (level III)
Amy	9	Both
Elisabeth	17	Hospital (level II)
Vanessa	18	Hospital (level II)
Holly	3	Both
Araminta	12	Primary Care
Alisa	7	Both
Eleonor	4	Both
Bianca	15	Both
**Focus Group C**		
**Name**	**Years Working**	**Hospital/Primary Care**
Barbara	37	Hospital (level III)
Angela	4	Hospital (level II)
Hugh	7	Hospital (level III)
Cameron	7	Primary Care
Malory	14	Both
Rita	13	Both
Rachel	4	Hospital (level III)

**Table 3 ijerph-18-12553-t003:** Interview script.

What does the concept of Obstetric Violence (OV) suggest to you? How would you define OV? Does the concept seem appropriate, or would you prefer other terms? Which? What relationship do you think there is—if any—between OV and gender violence? What do you think when you hear that there are patriarchal attitudes in childbirth care? Would you speak of systematic male chauvinism in delivery rooms? How do you feel, as professionals in childbirth care, when the word “violence” is used to talk about health work? What relationship would you establish between the concept of “humanization of care” and OV? From your point of view, are they different sides of the same coin or are they different things? Do you think that specifically classifying OV in the Criminal Code would help to prevent it and defend women’s rights or, on the contrary, would it contribute to making women vulnerable and victimizing them? Do you think that clinical practice is sometimes judicialized, leading professionals to exercise defensive medicine? How can OV be detected, defined, and punished if violence is a subjective concept? As midwives, do you feel that the medical hierarchy limits your autonomy? What kind of decisions would you like to be able to make, or what would you like to be able to influence more? What idea of motherhood do women in labor have? What social image do you consider that currently exists among women about childbirth and childbirth care? Do you consider that they have realistic information about the physiological process of childbirth? Do you agree with the fact that the medicalization of childbirth has caused a loss of knowledge and power in women? Do you consider that women are infantilized when they are in labor? What role does the partner or husband play in the prevention of OV? Is it sufficiently taken into account by health services? Can he or she also be a victim of the system and of OV? Do you think OV only occurs at the delivery room and at the time of delivery or that there are also bad practices in other areas (i.e., the administration desk) and times (i.e., pregnancy, puerperium)? Is there adequate emotional support for women in labor? Are women in labor sufficiently empowered regarding their health rights? Is there sufficient discussion about it with women? Do you agree with the concept of “shared decision-making”? Do you think that informed consent documents are adequate enough as they are currently done? How can they be improved?

**Table 4 ijerph-18-12553-t004:** Categories defined from the analysis of the midwives’ discourse.

Category	Examples
The maleficence of forgetting women’s vulnerability	“*The gynecological position is really a very vulnerable position for the woman, so then, well, the emotional repercussions that it has for this woman…professionals have to be very careful with it.*” “*To me, obstetric violence is…taking advantage, talking about the moment of childbirth, from the vulnerability of the naked woman.*”
Beneficence requires respect for women’s integrity and dignity	“*I would define it…so that an omission or over-intervention is made when being disrespectful with women and directly having an effect on their dignity as a person.*” *“It suggests to me an abuse to integrity.”*
Women’s autonomy is being removed from them	“*Women have their rights, authority, autonomy, and empowerment taken away. They are left defenseless because you have the truth.*”
A problem of social justice towards women	*“I don’t like to call it obstetric violence because I think that, in the end, I dread its getting restricted to obstetrics, I think it is violence towards all women…not only at this moment (childbirth), there are other moments: like contraception, sexuality, and adolescence in which we are also a bit marginalized…”*

## Data Availability

The data presented in this study are available on request from the corresponding author. The data are not publicly available due to privacy restrictions.

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
