# Peer review of "Obstetric Violence as an Infringement on Basic Bioethical Principles. Reflections Inspired by Focus Groups with Midwives"

_ijerph, 2021, doi:10.3390/ijerph182312553_

Round 1

Reviewer 1 Report

It is a very interesting study !

I think that this article contributes to the advancement of existing knowledge. Also, the quality of presentation is high and the research is well-designed and technically sound. I would recommend to accept this article in present form.  

Author Response

Thank you very much for assessing our manuscript so positively!

Reviewer 2 Report

The topic is interesting, but the paper has several red flaws in methodology and results. Introduction is not clear. Results provide references, is not an original text, discussion is missing... I think that the paper should be re-written again. 

Author Response

Thank your very much for your suggestions, that we have tried to include in the manuscript. Please, find our comments attached.

Reviewer 3 Report

See comments in attached pdf file.

Author Response

Dear reviewer,

Thank you very much for your positive assessment and suggestions.

Regarding the results, if you don't want to tell us how you would improve them more concretely, we will modify them according to other reviewers' suggestions.

Regarding the requirement of spell checking, our institution (University of Barcelona) had already paid for IJERPH's languag editing service before the manuscript was sent to reviewers. Therefore, and taking into account that we are not native speakers, we don't feel competent to judge your suggestions. We will get in touch with IJERPH's editors in order to check that you recieved the last version of the manuscript and, if you did, then we will discuss whether another language checking is required.

Thank you again!

Reviewer 4 Report

This paper entitled “Obstetric violence as the infringement of the basic bioethical principles. Reflections inspired by focus groups with midwives” presents a quite interesting topic, that deserves attention from academia and health settings. Overall, I think that the manuscript has good quality, although it can be strengthened. Therefore, I present my detailed comments and suggestions below.

In general, I have a major methodological concern: authors mentioned interchangeably focus group and in-depth interview, and this should be clearly clarified. In my opinion the main strengths of the current research are its coherence and approach, allying bioethical principles to midwives perceptions.  

I think that the title is too long and can be shortened.

Key-words: the selected key-words are ok.

The abstract is well written and structured, summarizing the main points of the work. However, authors applied abbreviations, which should be avoid or instead clearly defined in the first usage.

The Introduction comprises eight too short paragraphs, not being clear what is the main novelty of the work (or its contribution to the current state of the art). Additionally, authors presented mainly a conceptual introduction, being also relevant to include some empirical evidences. Again, authors applied abbreviations (such as UN, WHO) without specifying their meanings in the first usage and this should be corrected. In my opinion the presentation of the current state of art can be improved.

In the last paragraph, authors presented “the core objective” of the study and my question is: what are the secondary goals? I suggest being more specific in the presentation of the aims. Additionally, in lines 78-80, authors stated their conclusions and, in my opinion, it is too early to present this information, so I recommend that authors exclude the statement here.

In my opinion “Methods” section presents the main weaknesses that should be addressed by the authors. First, in the “2.1. Design” subsection authors mentioned simultaneously in-depth interviews and focus group and these are distinct strategies of data collection, which potentially provided distinguishable results. As far as I understood, authors performed three focus groups, applying a pre-established script. I do not understand why authors mentioned “interview” and this should be clarified. In the same subsection authors stated aiming to “understand their [midwives] experiences in obstetric violence” and it can be understood as a new objective that was not identified before. In my opinion it can be the secondary or specific goal (that was previously asked), but it should not be presented here but earlier. In the subsection “2.2. Context of the study” authors should provided more information about health-related variables, that is relevant to the topic under study (e.g., number of pregnancies, number of children per women, mean age of first pregnancy, etc.). Additionally, more details about the process of healthcare of Spanish women during pregnancy and delivery as well as expected roles of midwives should be added. Selection procedures and characteristics of participants are well described (subsection “2.3. Selection and characteristics of participants”. However, it is important to include information about how participants were allocated to groups. The subsection “2.4. Criteria for methodological rigor” is quite informative. In the subsection “2.5. Data analysis” authors should clearly stated the technique or procedure for data analysis, besides describing the steps. Moreover, I think that definitions of the North-American and European principles of bioethics should be placed in the introduction and not here. Lastly, I would like to know a little bit more about the codification: was it performed by independent researchers? What was the percentage of agreement between them? When they disagree, how consensus was reached?

In my opinion, section “3. Results and Discussion” authors presented the main findings and compared them with other studies. Moreover, authors provided some explanations or critical thinking about their results. Despite the main aims of this section were achieved, I would rather prefer a distinct structure: in this version of the manuscript authors tended to present the state of the art and then to present their findings. I think that an inverse order would work better: first present the findings and then discussed them. More than once, authors applied the word “grate” but I think it’s a lapse and the correct word would be “great”. I made an effort, but I cannot understand what authors want to say in the title of Table 3 when mentioned “prepare by the authors”. Concerning the name of categories, I have some doubts about the use of personal pronoun “my” since findings were based on midwives’ perceptions and not women themselves. In lines 407-409 authors included a sentence about suicide that is deeply out of context; I suggest to address it further or instead remove it.  

Authors presented some limitations and interesting suggestions for further research in the section “4. Research and future research lines”. However, I recommend adding and discussing two additional potential limitations, namely only one man was included in the focus groups and the self-selection of the participants. Additionally, considering that previous authors stated that finding could be generalize and then claimed they were not, I would like to read a further discussion about this issue.

I think that section “5. Conclusions” meets the expectations.

Authors provided an appropriated and updated list of “References”.

Good work and all the best with the paper!

Author Response

(The authors gave the same response as above.)

Round 2

Reviewer 2 Report

The paper has serious flaws, you mix results and discussion. There are sentences in the introduction have not relevant for the study. The methodology is not clear. You have not taken into account most of the comments from the first review. The paper need be rewritten. 

Author Response

Thank you very much for your suggestions. Please, see our comments in the attached file.

Faithfully,

Round 3

Reviewer 2 Report

Line 99: Correct the points and sentence. 

Line 113: Correct the phrase, lacking a point. 

Line 156: Correct the phrase. 

Correct table 4: Delete the column. I think it would be useful include units of meaning too. 

I don't think that you should combine results and discussion. I doubt about the originality of your paper. 

Author Response

Dear revisor,

We have been very carefully looking at the 3rd round suggestions from you and, unfortunately, we find it hardly difficult to amend them due to different reasons:

On the one hand, the suggestions referred to specific lines doesn’t make sense to us in our manuscript (I attach here) because of the following:

  • Line 99 corresponds to a section title.
  • Line 113 is a quote
  • Line 156 is a parenthesis referring to a table

We have checked older versions of the manuscript, but your suggestions don’t make sense either, as they refer to the same sentences listed above (title, quote…) or to deleted sentences.

So, could you please check that you have the same version of the manuscript than we?

On the other hand, you insist for the third time in two questions and that we consider it is difficult to amending:

1) Table 4: you asks us to include units of meaning, but this is not usually done in the qualitative manuscripts published in IJHERP.

2) Separating “Results” and “Discussion”. That can be a good idea but as we have already argued previously, when using qualitative methodology, the sections “Results” and “Discussion” can be unified in one section, in which case references can be included. In addition, the IJERPH’s instructions for authors allow this structure:  

  • Discussion: Authors should discuss the results and how they can be interpreted in perspective of previous studies and of the working hypotheses. The findings and their implications should be discussed in the broadest context possible and limitations of the work highlighted. Future research directions may also be mentioned. This section may be combined with Results. 

There are different examples of that in IJERPH that we suggested previously, such as: 

https://www.mdpi.com/1660-4601/18/2/404/htm and

https://www.mdpi.com/1660-4601/18/20/10610/htm  

Likewise, from our point of view we consider that this possibility of unifying both sections is enriching for potential readers. 

It is also important to stress that you are only one out of four revisors that doesn't like our article, as reviewer 1 accepted it for publication in round 1 and reviewers 3 and 4 accepted it in round 2, and they didn’t indicate anything in this sense.  

Can you accept the manuscript? It is important for us the acceptation.

Thank you very much for your understanding.

Yours faithfully,

Authors
